# Immune Response in Young Thoroughbred Racehorses under Training

**DOI:** 10.3390/ani10101809

**Published:** 2020-10-05

**Authors:** Katia Cappelli, Massimo Amadori, Samanta Mecocci, Arianna Miglio, Maria Teresa Antognoni, Elisabetta Razzuoli

**Affiliations:** 1Dipartimento di Medicina Veterinaria, Universià degli Studi di Perugia, 06126 Perugia, Italy; miglioarianna@libero.it (A.M.); maria.antognoni@unipg.it (M.T.A.); 2Centro di Ricerca sul Cavallo Sportivo, Universià degli Studi di Perugia, 06126 Perugia, Italy; 3Rete Nazionale di Immunologia Veterinaria (Italian Society of Veterinary Immunology), via Istria, 3, 25125 Brescia, Italy; m_amadori@fastwebnet.it; 4National Reference Center of Veterinary and Comparative Oncology (CEROVEC), Piazza Borgo Pila 39/24, 16129 Genoa, Italy; elisabetta.razzuoli@izsto.it

**Keywords:** thoroughbred racehorse, cytokines, immune response, stress, intense training

## Abstract

**Simple Summary:**

Stressful stimuli, both infectious and non-infectious, can modify and trigger an innate immune response and inflammation, via an attempt to restore a homeostatic state. Coping with stressors can be measured by different procedures, including the evaluation of immunological parameters. These are also modulated by exercise, which can be considered stress prototypic in the Thoroughbred racehorse. To evaluate the complex of physiological regulations during the training period, twenty-nine clinically healthy, two-year-old Thoroughbred racehorses were followed during their first 3 months of sprint training. Blood collection was performed at rest, three times until 90 days of training, for testing immunological parameters during incremental sprint training to evaluate its effect on the immunological status of the animals. During the training period, we observed the following: (A) an increase in red blood cell parameters that are crucial for exercise performance adaptation, improving O_2_ transport and muscle cell respiration; (B) variations of blood granulocytes; and (C) changes in inflammatory cytokine gene expression. On the basis of clinical and laboratory findings, training exercise probably played a major role in the modulation of the above parameters. These latter changes could be seen as a preparation of the innate immune system to respond quickly and adequately to environmental conditions.

**Abstract:**

Training has a great impact on the physiology of an athlete and, like all stressful stimuli, can trigger an innate immune response and inflammation, which is part of a wider coping strategy of the host to restore homeostasis. The Thoroughbred racehorse is a valid animal model to investigate these changes thanks to its homogeneous training and highly selected genetic background. The aim of this study was to investigate modifications of the innate immune response and inflammation in young untrained Thoroughbred racehorses during the first training season through haematological and molecular investigations. Twenty-nine Thoroughbred racehorses were followed during their incremental 3-month sprint exercise schedule. Blood collection was performed at time 0 (T0; before starting the intense training period), 30 days after T0 (T30), and 90 days after T0 (T90). Haematological parameters (red and white blood cells, haemoglobin, and platelets) were evaluated and haematocrit (HCT), mean corpuscular haemoglobin concentration (MCHC), and red cells width distribution + standard deviation (RDW-SD) were calculated. Moreover, via RT-qPCR, we investigated the expression of, Interleukin *1β (IL-1β),* Interleukin *4* (*IL-4)* Interleukin *6 (IL-6),* Interleukin *2 (IL-2),* Interleukin *3 (IL-3),* Interleukin *5 (IL-5)* Interleukin *8 (IL-8)*, Trasformig Growth Factor *β* and *α (TGF-β)*, Tumor necrosis factor *α* (*TNF-α)*, and Interferon *γ* (*IFN-γ*)genes. Main corpuscular volume (MCV) showed a significant (*p* = 0.008) increase at T90. Main corpuscular haemoglobin (MCH) and haemoglobin concentration (MCHC) values were significantly augmented at both T30 (*p* < 0.001) and T90 (*p* < 0.001). Basophils were significant increased at T30 (*p* = 0.02) and eosinophils were significantly increased at T90 (*p* = 0.03). Significant differences in gene expression were found for all the genes under study, with the exception of *IFN-γ* and *TNF-α*. In particular, *IL-2* (T30, *p* = 0.011; T90, *p* = 0.015), *IL-4* (T30, *p* = 0.009; T90, *p* < 0.001), and *IL-8* (T30, *p* < 0.001; T90, *p* < 0.001) genes were significantly upregulated at both T30 and T90 with respect to T0, *TGF-β* was intensely downregulated at T30 (*p* < 0.001), *IL-5* gene expression was significantly decreased at T90 (*p* = 0.001), while *IL-1β* (*p* = 0.005) and *IL-3* (*p* = 0.001) expression was strongly augmented at the same time. This study highlighted long-term adjustments of O_2_ transport capability that can be reasonably traced back to exercise adaptation. Moreover, the observed changes of granulocyte numbers and functions and inflammatory cytokine gene expression confirm a major role of the innate immune system in the response to the complex of stressful stimuli experienced during the training period.

## 1. Introduction

In order to optimize their interactions with their environment, humans and animals are prompted to adapt and mount a corrective response to noxious stimuli. Signal molecules activation for a prompt reaction can be induced by different types of stressors, both infectious and non-infectious, on the basis of the host’s responsiveness [1,2,3].

The innate immune system is a foundation of the host‘s coping strategies with both infectious and non-infectious stressors. Accordingly, a plethora of environmental, non-infectious stressors can induce danger signals, leading to innate immune responses, as usually observed after exposure to microbial stressors [4,5]. Hence, the immune system can provide biomarkers for monitoring animal health and welfare; such biomarkers can highlight the mechanisms leading to adaptive failure, abnormal behaviour, and poor welfare [6,7,8].

In this conceptual framework, stimuli like weaning [9,10], transport [11,12], pregnancy and lactation [2,13,14], environmental pollution [14], exercise [15,16,17], cancer, and psychic stress [1,2,18] can trigger an innate immune response and inflammation, via an attempt to restore a homeostatic state [19]. Accordingly, coping with such stressors can be evaluated by measuring immunological parameters [19].

Thoroughbred racehorses are sprinter animals, their muscles being highly adapted for sprinting; with training, horses can reach distances over 1000 m and speeds of 60–70 km/h [20]. This horse is a valid model to investigate changes in immunological parameters induced by training programs because of their intensive schedule and homogeneous background [17,21]. The effects of training are related to the type, duration, and intensity of the exercise; they can also include changes in immunological and haemato-biochemical parameters [17,22,23,24,25]. Thoroughbred racehorses are athletes characterized by physiological and anatomical adaptations to exercise. This horse, indeed, has been subjected to a careful genetic selection aimed at obtaining animals that, as athletes, must comply with physical, morphological, behavioural, and metabolic requirements in order to allow training and maximum performance in a race. In particular, muscle remodelling and blood modifications have been reported [26,27,28]. To plan a consistent training program for each horse, it is necessary to determine the training intensity, frequency, and duration, as well as the amount of work, rest, and recovery [29,30], in order to avoid overtraining syndrome [31,32,33,34].

Currently, it is known that a single and intense training session causes a much stronger innate immune response than a constant training workout; nevertheless, it has been noted that moderate training also causes an immune response [35,36]. In this respect, many studies have reported early modifications after exercise; these are characterized by an increase in haemato-biochemical parameters 5 mins after exercise and a return to basal levels after 60 mins [37,38,39,40]. However, few studies have investigated long-term changes over several weeks [21,41] during the first training period in Thoroughbred racehorses. Owing to the above, we hypothesized important modifications of circulating leukocytes and cytokine gene expression between 30 and 90 days after start of training. Indeed, in humans, many genes belonging to immune response pathways have been found to be modulated by exercise [42,43,44]. However, regarding this topic, insufficient data are available in racehorses [17,45]. Therefore, the aim of our study was to evaluate long-term immunological modifications during the first training period of Thoroughbred racehorses. To this purpose, we tested immunological parameters in horses at rest during the first 3 months of incremental sprint exercise training to evaluate its effect on the immunological status of the animals. Therefore, our working hypothesis implied the occurrence of physiological adjustments as a response to the complex of stressors experienced by young racehorses during their first training period, with a crucial role of the innate immune system.

## 2. Materials and Methods

### 2.1. Animals

Our study included nineteen 2-year-old Thoroughbred racehorses (7 males and 12 females; 350–450 kg, height 165–168 cm) that had never been trained for canter and gallop, which were housed in the Capannelle training center (Rome, Italy). The gorses were routinely treated for parasitic infections at 3-month intervals using ivermectin 2% (0.2 mg/kg) + praziquantel 25% (2.5 mg/kg) as oral paste. Also, all the animals were vaccinated against influenza and tetanus with current commercial vaccines and tested negative for antibody to equine infectious anaemia virus. In the 12 months before training started, the horses were managed similarly (individual housing, natural photoperiod, same alimentation, and natural indoor temperature); then, at the beginning of training, the animals were adapted to the new environmental conditions. The horses were fed three times a day with 2 kg of hay supplemented with green grass, 2 kg of mixed cereal concentrate (hay pellets, corn, oats, barley, and beans), with the addition of fruit and vegetables, and given water ad libitum. Semi-liquid mash was administered three times a week, which consisted of 1 L of olive oil and 1 L of honey and the addition of calcium lactate. During our study period, no vaccines were administered, and the animals showed good health conditions. They were checked at each sample collection through respiratory auscultation, heart exam, evaluation of rectal temperature, routine haematology, and clinical chemistry analyses.

Training was performed from Monday to Saturday at the same time for each horse under study. Samples were taken at rest on Wednesday halfway through the weekly workout, and the rest period from the previous workout was approximately 20 h. All animals were trained according to the same training schedule (Table 1).

Climatic conditions were defined by calculating the temperature humidity index (THI). This parameter is commonly used to quantify the heat stress on farm animals. THI values were calculated as described by Vitali and co-workers [46], using temperature and relative humidity data recorded at 12 AM at a weather station 6.6 km away from the Capannelle center (Table 2).

### 2.2. Sample Collection

The experimental period was divided into three times (April T0, May T30, and July T90) characterized by incremental training. Samples were collected before training and feeding once a month from April 2018 to July 2018, at 6:00 AM. April (T0) was the first month of training simulating competitions (gallop), whereas at T90 samples were collected before the race. Blood samples were taken from the jugular vein using Vacutainer tubes (10 mL; Terumo Corporation, BD brand; Tokyo, Japan) with Li-heparin and without additives by the veterinary practitioner in charge of assessing the health of the animals during the training season. Pharmacological treatments were not administered in the month before the first sample collection.

Buffy coat was recovered from 10 mL of total blood as previously described [21]. Briefly, heparinized blood samples were centrifuged at 2000 rpm for 10 min, and the layer of white cells was collected. The hypertonic solution Ammonium-Chloride-Potassium (ACK) (8.024 mg/L NH4Cl, 1.001 mg/L KHCO3, and 3.722 mg/L ethylenediamine tetraacetic acid-EDTA, pH 7.3) (1:4) was used to lyse red blood cells: samples were incubated for 10 min in ice and centrifuged at 2000× *g*. The resulting pellets were resuspended in 2 mL of TriZol reagent at room temperature (Thermo Fisher Scientific, Waltham, MA, USA) and stored at −80 °C until RNA extraction.

### 2.3. Haematological Parameters

Blood (10 mL) was collected from the jugular vein in a tube containing K3-EDTA (Venoject, Terumo, Italy). Analyses were performed within 3 hours from blood collection. A complete blood count (CBC), including leukocyte differential counts, was performed using a laser haematology analyser (Sysmex XT-2000iV; Sysmex, Kobe, Japan), validated on equine blood [1], and equipped with a multispecies software. This analyser combines laser-based flow cytometry and impedance technology. The following analytes were measured: white blood cells (WBCs), neutrophils, lymphocytes, monocytes, eosinophils, basophils, red blood cells (RBCs), mean corpuscular volume (MCV), haemoglobin (Hb), and platelets (PLTs). Haematocrit (HCT), mean corpuscular haemoglobin (MCH), mean corpuscular haemoglobin concentration (MCHC), and red distribution width standard deviation (RDW-SD) were calculated automatically by the analyser.

The Sysmex analyser has been recently validated for the leukocyte count in veterinary medicine [1,21,47].

### 2.4. RT-qPCR Analyses

Total RNA from T0, T30, and T90 samples was extracted using TriZolPlus RNA purification kit (Thermo Fisher Scientific, Waltham, MA, USA), according to the manufacturer’s directions. Therefore, RNA quality and quantity were evaluated using the NanoDrop1000 spectrophotometer (Thermo Fisher Scientific, Waltham, MA, USA) and electrophoresis in a denaturing 1.2% agarose gel (Thermo Fisher Scientific, Waltham, MA, USA). Each sample was reverse-transcribed using 500 μg of total RNA; the reaction was performed with the OneScript® cDNA Synthesis Kit (Applied Biological Materials Inc., Richmond, BC, Canada) as specified by the manufacturer. The accession numbers and primer sets utilized to evaluate gene expression are reported in Table 3. Succinate dehydrogenase complex subunit A *(SDHA*) and β2 microglobulin *(B2M*) were chosen as reference genes in agreement with a previous study [36] and tested with geNorm algorithm, included with the CFX maestro software (ver. 4.1 BioRad, Hercules, CA, USA).

geNorm provides a ranking of the tested genes by considering their expression stability and selecting reference genes according to the stability measure M (average pairwise variation of each gene against all the others) [48].

Protocols for interleukin-1β (*IL-1B*)*, IL-4,* and *IL-6* genes had been previously set up [21,49], while primers for *IL-2, IL-3, IL-5, IL-8,* transforming growth factor β1 (*TGFB1*)*,* tumour necrosis factor-α (*TNFA*), and interferon-γ (*IFNG*) were designed using the Primer-BLAST free software available on line on https://www.ncbi.nlm.nih.gov/tools/primer-blast/; these were placed at exon-exon junctions or at different exons to avoid biases due to genomic DNA amplification. Data are expressed as 2^−ΔΔCq^ ± 1 standard error [50].

### 2.5. Statistical Analysis

Raw data from CBC analysis were imported into R (ver. 3.4.1, https://www.r- project.org) for statistical analysis, as described previously [21]. The differences between groups were analysed by one-way ANOVA. After including a second independent variable (sex) in the statistical model, two-way ANOVA was applied to the neutrophils parameter, as it was the only one in which gender had shown a statistically significant effect. A post-hoc test using the “emmeans” R package (https://CRAN.R-project.org/package=emmeans) was applied to monitor the changes in the parameters under study over time, setting the significance at *p* < 0.05. RT-q PCR samples were divided into three groups (T0, T30, and T90) and the Shapiro–Wilk test was used to determine the likelihood that the expression values of the samples in a biological group were obtained from a normally distributed population. Then, modifications in the relative gene expression between groups were evaluated by one-way ANOVA. The threshold was set at *p* < 0.05. Data in the manuscript are expressed as means of fold change with the standard error using CFX maestro software (ver. 4.1- BioRad, Hercules, CA, USA). Tendencies were declared at *p* < 0.1.

### 2.6. Ethical Animal Research

Sampling was allowed by the Italian Horse Racing Board and performed by the authorized veterinary practitioner during routine controls to assess the health of the animals in the course of the training season. Prior to sample collection, written owner or trainer consent was obtained for each animal. Anyway, the animal care procedures were compliant with the European recommendations (Directive 2010/63 / EU) for the protection of animals used for scientific purposes.

## 3. Results

### 3.1. Haematological Parameters

All results are expressed as mean values ±1 standard error in Table 4. In particular, WBC, HCT, HGB, PLT, RDW-SD, and RDW-CV were not significantly modulated (*p* > 0.05) during our study. RBC values decreased at both T30 and T90 with respect to T0; however, variations were not significant, and we report only a tendency (*p* = 0.07) at T90. MCV showed an irregular trend characterized by a decrease at T30 and a significant (*p* = 0.0082) increase at T90. MCH and MCHC values were significantly augmented at both T30 (*p* < 0.001) and T90 (*p* < 0.001). Concerning the white blood cells, we observed an increase in neutrophils at all times of sampling with respect to T0, although this was highly influenced by gender; accordingly, a significantly lower mean value was observed in males compared to females (male 4.052 ± 0.35; female 5.01 ± 0.29), with a tendency of increased values in both groups at T90. Also, monocytes increased at T30 and T90 compared to T0, while lymphocytes initially showed an increase (T30; *p* = 0.08) followed by a decrease of this cell type at T90. Regarding basophils and eosinophils, both cell types were increased at T30 and T90. However, the increase was significant at T30 (*p* = 0.028) for basophils and T90 for eosinophils (*p* = 0.036). On the whole, the prevalence of band neutrophils was minimal and within the reference values for horses in all the animals under study and at all time points (data not shown).

### 3.2. RT-qPCR

Figure 1 shows the relative expression of *IFNG*, *TNFA*, *IL-1B*, *IL-2*, *IL-3*, *IL-4*, *IL-8*, and *TGFB* genes of leucocytes obtained from 19 horses grouped by time of training. Data were normalized using two reference housekeeping genes (*SDHA* and *B2M*); both genes displayed a relatively high stability with M values of 0.7, far below the accepted limit of 1.5.

Significant differences in gene expression were found for all genes under study, with the exception of *IFNG* and *TNFA*. In particular, *IL-2* (T30 *p* = 0.011; T90 *p* = 0.015), *IL-4* (T30 *p* = 0.009; T90 *p* < 0.001), and *IL-8* (T30 *p* < 0.01; T90 *p* < 0.01) genes were significantly upregulated (Figure 1) at both T30 and T90 with respect to T0. *TGFB* was intensely downregulated at T30 (*p* < 0.01). *IL-5* gene expression decreased at T90 (*p* < 0.01), while *IL-1B* (*p* = 0.005) and *IL-3* (*p* = 0.001) expression was strongly augmented at the same time.

## 4. Discussion

Moderate exercise determines a boost effect on immune functions, whereas high-intensity training causes reduction of immune functions [51], which may lead to outright overtraining syndrome [33,34,52,53]. Moreover, young horses are more sensitive to exercise-induced immune suppression [54,55,56] and the training should be arranged in terms of duration and intensity in order to stimulate the immune system without reaching the above immunosuppressive state. In this study, physiological changes were monitored in young Thoroughbred racehorses during their first training period, with respect to the immunological status of the animals.

Acute exercise has an effect on white blood cells and cytokines [53]; the effect of training is less known, especially in horses, and we hypothesized that it should mimic, if well planned, the positive effects of prolonged moderate exercise [57,58].

For these reasons, it is important to monitor changes in the immune system during training in order to set optimal protocols that allow for better performances, while preserving animal welfare and immune system homeostasis. The animals under study, observed over three months, allowed us to detect a gradual adaptation to the new conditions.

This study highlighted significant changes in the time-course of leukocytes, red blood cells, and cytokine gene expression. In particular, RT-qPCR analysis shows modulation of *IL-1B*, *IL-2*, *IL-3*, *IL-4*, *IL-5*, *IL-8*, and *TGF**B1* gene expression at different time points. Regarding leukocytes, we evaluated the trend of monocytes, lymphocytes, and granulocytes. The latter are classified into three groups, basophils, neutrophils, and eosinophils, on the basis of the properties of their granules. Concerning the trend of neutrophils, we observed an increase of these cells at each time of sampling, with higher levels 90 days after the start of training. Since the prevalence of band neutrophils was always negligible, we can reasonably rule out infectious agents underlying the observed neutrophilia [59,60], which is reminiscent instead of a stress leukogram due to endogenous corticosteroid release [58]. Monocytes and eosinophils showed the same trend of neutrophils, with a significant increase at 90 days for eosinophils (*p* < 0.05), at 30 days for basophils (*p* < 0.05), and a tendency of increase (*p* < 0.1) for monocytes and lymphocytes followed by a decrease and a return to starting values after a further 60 days (T90).

One explanation of this trend in leukocytes can be cortisol secretion in response to exercise. In addition to that, catecholamines such as epinephrine cause leukocytes to detach from the vascular wall and microvasculature into the main circulation; this, as well as splenic contraction, may significantly elevate white blood cell counts [58]. Indeed, young horses are characterized by higher blood cortisol levels post-exercise compared to older animals and those with greater training experience [51,61]. Further studies on the expression levels of L-selectin in granulocytes will help characterize this haematological feature, in agreement with an established model in cattle [62]. Interestingly, highly trained human subjects show mild hypercortisolism that may be an adaptive change to chronic exercise [63]. The changes in red blood cells should be interpreted as the adaptation to exercise of trained horses improving O_2_ transport. Accordingly, O_2_ content in blood varies with Hb concentration at any given O_2_ partial pressure; this is of paramount importance for exercising muscle cells releasing H+, CO_2_, and lactate, all of them causing a rapid decrease of Hb-O_2_ affinity [64].

In this study, we also tested the gene expression of important cytokines involved in the immune response. We decided to test the trends of *IL-1B*, *IL-2*, *IL-3*, *IL-4*, *IL-5*, *IL-8, TNFA, IFNG*, and *TGFB1* genes involved in inflammatory response and leukocytes activation, which have already been tested in previous studies [57,65,66]. Seven cytokine genes out of nine were modulated during training. The expression of *IL-1B*, a pro-inflammatory cytokine pivotal to innate immune response, was stable between T0 and T30, then we observed a significant increase at T90. IL-1B is produced by many cell types, in particular monocytes, neutrophils, and eosinophils. Also, *IL-3* gene expression showed the same trend as *IL-1B*. IL-3 is a very important cytokine that induces differentiation of hematopoietic stem cells in myeloid progenitor cells and has a pivotal role in the activation of monocytes, neutrophils, and eosinophils [67]. In accordance with our findings, these cell types were increased at all times of observation. IL-2 is a type I cytokine produced mainly by Cluster of differentiation 4 (CD4+) cells and in minor amounts by Cluster of differentiation 8 (CD8+) cells. This cytokine functions as a growth factor for a wide range of leukocytes, including T, B, and NK cells, but it can also exert functional effects on neutrophils [68]. In our study, we observed the upregulation of this gene expression at day 30 of training, which was associated with an increase in circulating basophils and lymphocytes. This data is in agreement with a recent study that suggests the activation of basophils by IL-2 [69]. Also, *IL-8* and *IL-4* showed the same trend as *IL-2*. *IL-8* is expressed by many cellular types, such as monocytes, endothelial cells, and fibroblasts [70]. This chemokine acts as a potent chemoattractant for granulocytes, inducing an increase in neutrophils, eosinophils, and monocytes. Indeed, *IL-8* expression was associated with an increased prevalence of these cell types.

Concerning *IL-5,* we observed a constant decrease of its gene expression during our experiment, with a significant decrease at 90 days of training. IL-5 is expressed in T helper (Th2) responses and is associated with the activation of Th2 cells and eosinophils; in particular, IL-5 is involved in the recruitment of eosinophilic precursors in blood [71,72]. In our study, a decrease in *IL-5* was associated with an increase of *IL-3* gene expression and circulating eosinophils. Moreover, the *IL-4* increase during the training period could be due to the activation of eosinophils and basophils. IL-4 is a cytokine that is able to induce T-cell differentiation and Th2 polarization, acting as a regulator of the immune response [73].

TGFB1 is a member of the TGFB superfamily that includes three isoforms (TGFB1, 2, and 3). TGFB1 has a pivotal role in controlling the immune response and shows different activities on different cellular types. Most leukocytes are able to secrete TGFB1 [74]. This molecule has predominantly suppressive effects on macrophages and monocytes; indeed, it can inhibit their activation and prevent the production of reactive nitrogen and oxygen species. In agreement with this concept, in our study, *TGFB1* gene expression showed a trend opposed to those of basophils and monocytes. Indeed, we observed the downregulation of *TGFB1* 30 days after the start of training, while basal levels were restored after 90 days of exercise.

Unfortunately, we were unable to compare the trend of the same parameters in a group of untrained, age-matched, control subjects, due to the current practices with racehorses, especially in Thoroughbreds. Indeed, foals begin their training season at two years of age and there were no subjects of the same age who were kept untrained on the same farm. In a previous study, Cappelli and co-workers showed that the basal levels of immune-related genes were significantly higher in athletic horses (adult thoroughbred highly trained) compared to sedentary animals (mares at the end of their racing career, and out of training programs for at least 1 year) [17]; yet, in our experimental design there was too strong of a difference of age and physiology between young foals and mares to allow for a meaningful comparison.

We reasoned that training had undoubtedly had an important effect on the parameters under study, although we cannot completely rule out possible effects of other environmental stressors. Among the possible side stressors, heat might have affected the T90 values observed in July. Such a stressor deserves particular attention, since it can induce dramatic cytokine responses in farm animals [75]. Yet, on the basis of further checks on the THI data shown in Table 2, we observed no single day with THI min > 70 in the 15 days before the July sampling. This means that horses could adequately rest at night and probably get over the heat stress experienced in the daytime, as shown in established models of heat stress in farm animals [46]. In addition to that, blood sampling was always carried out around 6 AM (i.e., at a time which is not associated with acute heat stress). On the whole, the effects of heat stress were probably negligible in the time period under study. As for other environmental stressors, diet and housing conditions were kept homogeneous for all of the animals; these were maintained under strict control by the farm operators during the samplings and over one year before the training period.

Finally, there is strong evidence of a major post-transcriptional regulation of cytokine responses, mainly based on mRNA stability, and 3’ untranslated regions of mRNAs are a key target of such post-transcriptional control actions [76]. Accordingly, changes in cytokine gene expression should be viewed in terms of the alertness and preparedness of the innate immune system. This would suggest that the changes allowed the horses to be poised to better react to diverse environmental stressors.

On the whole, the observed increases in *IL-1B* gene expression in blood cells are in agreement with previous results from adult trained horses shortly after exercise; these also showed significant changes in *IFNG* and *TNFA* genes [65], which were not observed in our model of a long-term training regimen. Our results show that changes in gene expressions should be viewed as a peculiar long-term adjustment, which may be reasonably associated to improved functions of granulocytes [66], which also increased over the training period.

## 5. Conclusions

Our findings depict a peculiar coping strategy of young Thoroughbred horses to training for their first racing season. In this operational framework, the long-term adjustments of O_2_ transport capability and granulocyte number and functions played a fundamental role. The observed changes in inflammatory cytokine gene expression confirm a major role of the innate immune system in forging the environmental fitness of young horses after the completion of their training program.

## Figures and Tables

**Figure 1 animals-10-01809-f001:**
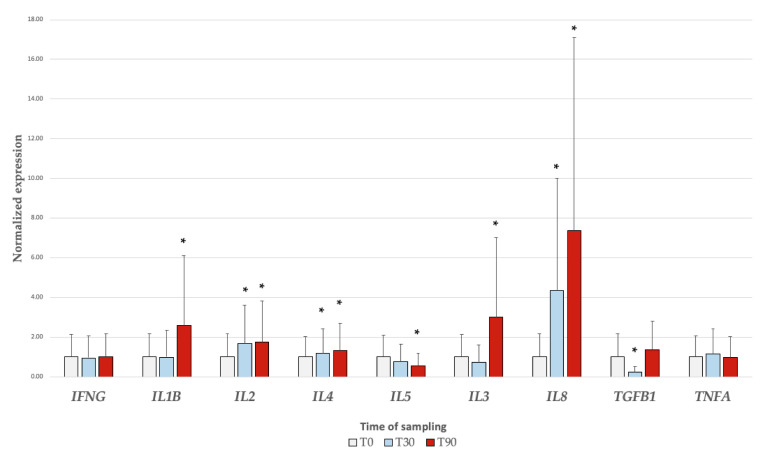
Results of cytokine gene expression. Data are expressed as 2^−ΔΔCq^ ± 1 standard error. * indicates *p* < 0.05 with respect to T0 as shown by ANOVA. *IFNG*: interferon γ; *IL-1β*: interleukin-1β; *IL-2*: interleukin-2; *IL-4*: interleukin-4; *IL-5*: interleukin-5; *IL-3*: interleukin-3; *IL-8*: interleukin-8; *TGFB1*: transforming growth factor β1; *TNFA*: tumour necrosis factor α.

**Table 1 animals-10-01809-t001:** Daily program completed by each animal involved in the study. Speeds: Walk, 100 m/min. Trot 200 m/min. Canter, 350 m/min. Gallop, 1000 m/min (min: minutes).

April T0	May T30	July T90
15 min Walk	15 min Walk	15 min Walk
10 min Trot	10 min Trot	10 min Trot
6 min Canter	6 min Canter	6 min Canter
Tuesday: 1 min Gallop	Tuesday: 2 min Gallop	Tuesday: 4 min Gallop

**Table 2 animals-10-01809-t002:** Environmental conditions during the study. Results are expressed as monthly mean value ± 1 standard deviation. THI values were evaluated as previously described [46]. THI: temperature humidity index; T: temperature; UR: relative humidity; min: minimum; max: maximum.

Period	T-max °C	T-min °C	UR-max %	UR-min %	THI-max	THI-min
T0 (April 2018)	24.7 ± 3.7	6.0 ± 4.9	97 ± 0.1	37 ± 0.1	76.2 ± 6.0	48.1 ± 6.2
T30 (May 2018)	30.2 ± 7.3	10.7 ± 4.4	95 ± 0.1	44 ± 0.1	85.6 ± 12.1	53.3 ± 5.9
T90 (July 2018)	38.2 ± 3.7	19.9 ± 1.8	83 ± 0.1	29 ± 0.1	99.1 ± 5.3	64.0 ± 1.8

**Table 3 animals-10-01809-t003:** Sequences of primers employed in this study specific for the following genes: transforming growth factor β1 (*TGFB1*), interleukin-1β (*IL-1β*), tumour necrosis factor α (*TNFA*), interleukin-8 (*IL-8*), interleukin-5 (*IL-5*), interferon γ (*IFNG*), interleukin-4 (*IL-4*), interleukin-2 (*IL-2*), interleukin-3 (*IL-3*), succinate dehydrogenase complex subunit A (SDHA), and β2 microglobulin (B2M).

Gene	Primer Forward	Primer Reverse	Amplicon Length	Accession
*TGFB1*	CGGAATGGCTGTCCTTTGATG	CCCACGCGGAGTGTGTTAT	127	NM_001081849.1
*IL-1B*	TGATGCAGCTGTGCATTCAGT	GCACAAAGCTCATGCAGAACA	146	NM_001082526.1
*TNFA*	AGCCTCTTCTCCTTCCTCCTT	CAGAGGGTTGATTGACTGGAA	123	NM_001081819.2
*IL-8*	CTGGCTGTGGCTCTCTTG	CAGTTTGGGATTGAAAGGTTTG	133	NM_001083951.2
*IL-5*	ACCTGATGATTCCTACTCCTGA	CCCCTTGGACAGTTTGATTCT	99	NM_001082499.1
*IFNG*	GCTGTGTGCGATTTTGGGT	ATCCAGGAAAAGAGGCCCAC	130	NM_001081949
*IL-4*	AAGAATGCCTGAGCGGACTG	TGGCTTCATTCACAGTACAGCA	75	NM_001082519.1
*I-L2*	GAAGAAGAACTCAAACCTCTG	TTCCTGTCTCATCATCATATTC	148	NM_001085433.2
*IL-3*	TGAAGGATCTAAACACGACACC	CCTTGAAACTAGGGACAGCTC	96	JL628807
*B2M*	TCCTGCTCGGGCTACTCTC	TGCTGGGTGACGTGAGTAAA	83	NM_001082502.3
*SDHA*	GCGCGCTTCAGACGATTTAT	CCAGTGCTCCTCAAATGGCT	146	XM_014734954.2

**Table 4 animals-10-01809-t004:** Haematological parameters. Data are expressed as mean values ± 1 standard error.

Parameter	Reference Range2-year-old Thoroughbred Horses in Training ^∞^	Laboratory Established Reference Range for Horses [1]	April T0	May T30	July T90
WBC (×10^12^/L)	7.3–12.7	6.0–12.0	9.59 ± 0.37	10.11 ± 0.34	10.12 ± 0.34
RBC (×10^12^/L)	8.7–11.7	8.6–12.0	10.29 ± 0.18	10.01 ± 0.19	** 9.92 ± 0.19
HGB (g/dL)	12.8–16.6	11.5–18.0	14.67 ± 0.37	14.70 ± 0.51	14.23 ± 0.51
HCT (%)	34–45	35–46	37.59 ± 0.63	36.53 ± 0.65	37.59 ± 0.65
PLT (×10^9^/L)	127–206	200–450	152.63 ±11.02	162.26 ± 9.63	145.68 ± 9.63
RDW-SD	NA	NA	33.64 ± 0.41	32.78 ± 0.52	34.23 ± 0.52
RDV-CV	24.0–27.0	23–27	27.49 ± 0.38	27.41 ± 0.27	27.43 ± 0.27
MCV (fL)	37.0–42.1	41–49	36.71 ± 0.57	36.49 ± 0.17	* 37.19 ± 0.17
MCH (pg)	13.7–15.7	12.8–14.1	14.31 ± 0.16	* 14.69 ± 0.07	* 14.86 ± 0.07
MCHC (g/dL)	35.9–37.9	34–38	39.05 ± 0.28	* 40.27 ± 0.21	* 40.03 ± 0.21
Neutrophils (×10^12^/L)	4.0–6.0	2.7–6.7	4.66 ± 0.28	4.80 ± 0.33	** 5.27 ± 0.33
Eosinophils (×10^12^/L)	0–0.3	0.1–0.6	0.17 ± 0.04	0.23 ± 0.05	* 0.27 ± 0.05
Basophils (×10^12^/L)	0–0.2	0-0.2	0.026 ± 0.03	* 0.033 ± 0.03	0.026 ± 0.03
Lymphocytes (×10^12^/L)	2.7–4.4	1.5–5.4	4.29 ± 0.23	** 4.57 ± 0.16	4.05 ± 0.16
Monocytes (×10^12^/L)	0.26–0.56	0.1–0.2	0.45 ± 0.03	0.48 ± 0.03	** 0.49 ± 0.03

NA: not available. * indicates a significant difference assessed by ANOVA test with respect to T0 (*p* < 0.05). ** indicates a tendency (0.1 < *p* > 0.5) with respect to T0. Reference ranges are given in agreement with an established dataset (^∞^
https://www.rossdales.com/laboratories/reference-ranges) and published Reference Intervals (RIs) [1].

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
