# Peer review of "Immune Response in Young Thoroughbred Racehorses under Training"

_animals, 2020, doi:10.3390/ani10101809_

Round 1

Reviewer 1 Report

The authors have drawn together a fascinating literature in support of their study and observations, and in the process have necessarily crossed several discipline boundaries, both traditional and new. Illustration of the relatedness and relevance of these different subject areas in a study that has the potential to demonstrate the applied relevance and practical significance of the mechanisms described for the benefit of those responsible for the daily care of the racehorse is a very worthwhile exercise. Anything reliable and justifiable that we can do in support of maintaining the health and welfare of the racehorse has my support, but it needs to be evidence-based. In this regard, I'm not sure this paper achieves its objectives. Because of methodological deficiencies in the execution of the study, presentation and dependence on concepts that are either inadequately developed in the horse or poorly supported by the literature cited, over interpretation of results, and presence of ambiguities introduced by issues of language and terminology, the authors do not reach their target, which they state as being "…to investigate, for the first time, the long-term immunological modifications in young untrained racehorses during the first training season ". While they do identify a number of limitations in their investigation, they do not appropriately apply this understanding to their results and conclusions. The paper also relies heavily on a literature published by the authors and their co-workers, and they are thus frequently citing themselves. To some degree this may be inevitable, but in the interests of objectivity and impartiality it is highly desirable that a broader literature be consulted.

In this article, aimed at a more general and applied audience, there needs to be some explanation of what is meant by inflammation and immunological status. These may to some degree be issues of language, but are the authors adopting a term used generically in immunology to describe a heightened level of response or activation of the immune system or are they referring to an actual whole-body clinical inflammatory response or are they referring to localised inflammatory responses? Is the experimental treatment activating the immune system or mechanisms that are common to immune function? Also, the authors might define what they mean by an "immune response". Do they really mean an immune response (to an antigen, for example), or do they mean "a response by the immune system", or do they mean a change/response in mechanisms traditionally or ordinarily characterised as part of the immune system but in fact shared with a much broader range of body system responses? Since they are applying their discipline interests in a very applied sector, the racing industry, and since participants in the industry have a tendency to grasp at straws when looking for a performance advantage, these clarifications are not merely scientific dogmatism.

The authors appear to be using the term "adaptation" very loosely and often inappropriately. Adaptation is part of an evolutionary response to better fit a species to its environment. If, as in the present case, they are referring to a short term (non-evolutionary) response within an individual they should use the words "acclimatisation" or training response. The capacity within an individual to mount a response to training is an illustration of the species' flexibility and illustrates the results of evolutionary adaptation in action, but the training response itself is not adaptation. Confusion regarding these concepts is a recurrent problem in this paper, and methodological issues aside, represents the greatest weakness.

Although I recognise the idea is used similarly elsewhere in the literature, I am not comfortable with the idea of three months representing long-term training. Even in Thoroughbred racehorses training continues for 2-3 years. The authors and I may not be able to rewrite published literature, but I think it is incumbent upon authors to address terms and their use critically rather than simply adopt them, especially when they are central to the question under study. In this investigation in particular, the issue of the length of the training period is pivotal since the group is small, results are equivocal, and the authors make frequent reference to acute as well as sustained responses.

There is an extensive literature relating to the response of the equine erythron to training to which the authors have made limited reference. Reference to practice normal values for a UK veterinary practice is not sufficient unless that laboratory performed all the blood work for this study - does the laboratory that performed the haematology not have its own reference values? How did the authors confirm the equivalence of laboratory procedures, or did they establish their own reference ranges, and if so, how?

Nothing in the data presented confirms that the responses seen are training responses. It would seem unlikely that training had nothing to do with the changes observed, but this study does not prove this. Numerous changes took place over this period including change in location, exposures that the authors did not measure, increasing age, the possibility of intercurrent disease, changes in diet from the home environment. And in addition, no allowance appears to have been made for the time course or possible persistence of gene activation or of the products of that activation. At the very least, acknowledging the breadth of the audience that may read these findings, some guidance on the significance of these factors would be helpful.

The paper suffers from significant problems with use of language, such that statements tend to be very complex and sometimes ambiguous, and often difficult to understand. Inappropriate use of the word "adaptation" may perhaps reflect this issue of language, as may other issues.

The paper uses short forms that are not defined (for example, UR-max).

Line 28 - Suggestion: …like all stressful stimuli, "evokes responses in existing physiologic mechanisms to more fully utilise the functional range of that and of dependent mechanisms so as to support homeostasis". [The flexibility the organism exhibits by being able to recruit and augment function in existing mechanisms reflects species adaptation. Breeding (a.k.a. in domestic animals) can be used to enhance the degree of flexibility or functional range of these mechanisms - a process which in the natural world is achieved through natural selection and is called "adaptation".]

Line 74 - these results are uncontrolled. Unless horses are chosen to 1st race at two years of age by random selection, the benefit of starting at two cannot be reliably determined. Many horses first start racing at older ages because they are not sufficiently well developed to start racing, or more importantly, because they have health issues that delay training. The effect may thus be one of selection rather than a specific benefit of early training.

Line 99 - What is this poor performance syndrome to which reference is made, the statement is not supported by any equine reference. Is this a well-defined entity in the horse? Similar concerns apply to the idea of "over training" - has this been characterised in the horse?

Line 132 - "mean" corpuscular hemoglobin?

Line 191 - this material cannot be used as a supporting citation if it cannot be accessed by a reader.

Line 211 - these results are definitely not "stepwise", this is a misrepresentation

Line 213 - since the paper is written in English, the authors need to be cautious in the terminology they use since interpretation for English-speaking jurisdictions may be different. As an example, Thoroughbred racehorses are not "sport horses", the latter refers to horses such as showjumpers and event horses.

Line 216 - as an example of the danger of using a fairly narrow literature source, the authors describe the mare as not being a suitable control for their experimental animals, then proceed to cite a reference that did just that. We can't have it both ways! On line 220 the authors state that "… training has undoubtedly an important effect on the parameters under study", yet in the next sentence acknowledge there are doubts, yet in their conclusions and abstract they make firm statements concerning the significance of their findings. I suggest the authors spend more time addressing the time course of changes in mRNA and gene activation and consider investigating ways of examining the parameters under study over time rather than examining levels at a single point in time.

Line 244 - this statement is confused, plus reference should be to oxygen carrying capacity and aerobic exercise.

Line 247 - not so, this is only partially demonstrated, results are inconsistent, uncontrolled, and subject to methodological limitations. And only some changes were significant, while the direction of these changes was not systematic and was most certainly not stepwise as stated earlier in the paper. The authors describe another reason to view the results with caution in the paragraph starting at line 250, but choose to ignore these limitations in stating the significance of their results. The authors introduce in this paragraph the potential for acute responses in cytokines to be observed in relation to exercise - what would the time course of these changes look like and were any controls applied in the experimental protocol to deal with this issue?

Line 268. This study absolutely does not justify these conclusions, which succinctly reiterate why an appropriately designed study to investigate these mechanisms might be very worthwhile.

Line 268 - we may once again be dealing with inappropriate word use stemming from use of language, but as stated, these conclusions are not supported by the the results presented or by the reliability of the experimental methodology.

Reviewer 2 Report

Line 2 Either insert "the" as in "the Thoroughbred racehorse" or change "racehorse" to the plural "racehorses."

Line 17 "the" Thoroughbred racehorse

Line 20 "months of" not "month"

Line 35 and throughout The measurement intervals are not really monthly. They are 0, 30, and 90 days of training, whereas monthly would include 60 day measurements as well. Please adjust throughout the manuscript.

Lines 39-42 No results are stated in the abstract. Just gross overall impressions. Please be more specific with results here.

Line 54 "sprinters"

Line 57 Might consider use of the following references:

Foreman JH, Bayly WM, Grant BD, Gollnick PD. Standardized exercise test and daily heart rate responses of Thoroughbred horses undergoing conventional race training and detraining. Am J Vet Res 1990;50:914‑920.

Foreman JH, Bayly WM, Allen JR, Matoba H, Grant BD, Gollnick PD. Muscle responses of Thoroughbred horses to conventional race training and detraining. Am J Vet Res 1990;50:909‑913.

Line 72 Reference 29 is not Thoroughbred racehorses.

Line 93 "administered"

Line 106 Table 2 The column labelled "T-mim" should be "T-min"

Line 116 "samples" should be "sample"

Lines 250-266 This paragraph is perhaps the most critical of the manuscript and should be expanded. This reviewer would appreciate more discussion of the significance, or lack thereof, of the changes seen in ILs in this study. That is the crux of the matter...why did they change, and what is the significance of the changes? What do the changes mean for the horse in training, and moving forward toward racing? Just stress? or is something else going on physiologically? 

Reviewer 3 Report

This study documented the changes in various immunological, hematological, and mRNA cytokine markers over the course of an initial training phase in young, untrained Thoroughbred racehorses. The investigators observed changes in some hematological parameters as well as mRNA cytokine gene expression markers over the training period. 

The manuscript and its findings, while interesting, requires some majors edits before serious consideration of publication. 

General Comments/Minor Edits

Authors should invest in manuscript editing service; needs revising from grammatical standpoint/word choice in certain instances (e.g. Line 28: athlete; Line 47: "danger signals"; Line 49: "host’s perception"; Line 67: "Nowadays")

Line 2: racehorses (plural)

Line 17: “the” Thoroughbred racehorse…

Line 80: previously untrained young horses?

Line 105: Replace ‘Table’ with ‘Trot’

Abstract

Needs major revision

Aims and Hypotheses need to be explicitly stated, currently they are absent

Needs to include main findings of the study with p-values, these are also curiously missing

Introduction

No clear hypothesis; needs to be reworded; it is not clear what “we hypothesized important modifications as regards the immune response…” means; need to overhaul this section of the manuscript to make clear what the actual aims and testable hypotheses are.

Methods

If possible, authors should provide more information about the animals, perhaps by including more on the diet of the animals during training and if body weights or composition measures or condition scoring were completed? Would provide a better picture of whole body physiological status of the horses prior to and following the training period.   

No age-matched control group; authors correctly point this out in their discussion; very big limitation in study design and limit veracity of findings

Need to specify when blood samples were collected relative to previous exercise bout

For gene expression, which 'housekeeping' gene are targets normalized to? SDHA?; If so, how was it selected? Does it change expression with training? Should be explicitly stated and reference or data provided to justify its use (include reference in Methods section).

Objective measure of final training status? Body weights, composition, BCS? Were all animals trained to the same physiological state? If yes, how can you justify this?

Results

Why are selected results from Table 4 also provided in Figure 1? Redundant

Figure 1. Y-axes need to be labeled (units)

Authors should show individual data points if possible to allow for transparency in the data set; also to distinguish males from females since authors allude to sex differences (bias) in discussion

Could large variations in gene expression be due to differences in training status between individual horses? Need more objective measures of training state

Figure/table legends need to define acronyms; should include language/description about the statistical test used to produce the p-values

Discussion

Should state main findings up front and place into context of previous literature (paragraph 2 should be paragraph 1); while the authors correctly identify their limitation of no control group, this could be moved to a separate section that outlines the study’s limitations

Greater discussion on the gene expression results is needed; 7 of the 9 targets showed changes over the training period however there is little discussion as to why and what their roles are or might be with regard to the adaptive response to training; why were these targets chosen?  

Round 2

Reviewer 1 Report

Animals-912184 - revised

I thank the authors for the efforts they have made to revise their manuscript and more fully explain and substantiate some of the concepts they use. I continue to have significant problems with this study, however, while some changes have raised other issues. The experimental subjects are a convenience sample and data were gathered under real-world circumstances, circumstances which always require compromise. The findings should be stated in a commensurate manner, with due circumspection. It would also be of value for the authors to consider to which particular audience they wish to speak. My specific issues are laid out below

In both the summary and the abstract the authors use the expression "innate immune response and inflammation". I strongly recommend removal of the words "and inflammation”. Notwithstanding their protestations concerning the accepted validity of "exercise immunology”, I see no value in promoting the notion that there is an inflammatory response associated with training, while I do see huge potential for misinterpretation, including the liberal application of anti-inflammatory agents where none are needed!

An animal may adapt by training but this is not adaptation in the evolutionary sense. The fact this word has been used previously in the literature supporting the authors' discipline doesn't make it right, it has been an inappropriate use of the term from the start unless appropriately qualified. We can't change published literature, but perhaps we can strike a blow for correct use of terminology by at least clarifying the way in which the term is used here and the discrepancy with the broader biological literature, and by better terminology, such as “adaptation to training” or “training adaptation”.

With regard to reference values for the blood work, the absence of adequate laboratory reference values for the population under study remains problematic, and is heightened by huge differences between the laboratory reference ranges and the values for the study horses. There are some quite startling differences, with indications that some study horses may have been dehydrated. Also, depending on the use of URL's in a published paper is dangerous because such addresses change frequently. One way around this problem might be to seek permission to include a copy of these values as a supplementary item to the paper. Regardless, the authors need to explain why their results differ so significantly from those in the reference ranges they use. The manuscript suggests some blood samples may have been taken during exercise (see below), making the basis for comparison questionable if the reference samples are resting samples.

I do indeed disagree with reference 30 in the first version of the manuscript. The cited study did not investigate this relationship but simply quoted other papers. Those papers, in turn, similarly failed to address the question of selection bias, with possible problems in those horses not selected being ignored. Statements made by the present authors reveal problems with superficial treatment of the literature - read more thoroughly and modify this statement accordingly, please. The extensive observations in the response to the reviewer on this point concerning the structure of the thoroughbred industry are appreciated but unnecessary. What is usual is not necessarily "best".

With regard to poor performance syndrome and overtraining, the authors may again be taking a somewhat simplistic view of the literature. The "poor performance syndrome" is not a syndrome, it is simply a horse not performing up to the owner/trainer's expectations. The causes can indeed be multitudinous and very real, but characterisation of such cases as a syndrome is overreach and above all else, does not adequately accommodate the possibility that what the owner wishes to characterise as a clinical problem may simply be unreasonable expectation. The unquestioning adoption of this term by the authors is not consistent with the scientific rigour that they presumably wish to project for their work. See note below regarding line 124. The same can be said of the term "overtraining". Despite the bibliography presented, the condition remains inadequately defined for the horse. With the sole exception of hypervolemia/red cell volume, hard data remain lacking. The paucity of equine literature on this topic is telling! Better that we not present either as well-defined conditions.

Line 124. These were horses in training - how on earth would you characterise a performance as indicating poor performance? What signs would you look for? There is no basis for this statement.

Line 216, original manuscript - now I see the ambiguity. If the word "nevertheless" is removed the statement makes sense - another language issue.

Line 40. "Mean" corpuscular hemoglobin concentration

P values - three significant digits would be quite sufficient - so many significant digits exaggerates the meaning of p values.

Line 122. Does this mean that these were exercise blood samples? How do the statements made on this line encompass the statements made on line 136, they seem to be contradictory. A very important piece of information to make clear!

Line 141. Does this mean treatments may have been administered before other collection periods? Were the animals truly treated as experimental subjects or were the authors simply given permission to sample periodically during a routine training period? If the latter, the sample must be characterised as a convenience sample. Were any of these horses given hematinics?

Line 268. This interpretation may hold for the neutrophils, but it doesn't hold for the lymphocytes, which should decrease but which in fact increased in relation to the laboratory reference values. Do the authors have an explanation?

Line 355. The authors continue to draw conclusions that are not adequately supported by their findings. The results are far too variable and inconsistent, while the authors continue to provide only incomplete interpretations. There is no guidance on reliability of the measures made of cytokines, such as how inherently variable they may be from sampling to sampling. The findings are interesting, but they do not justify the conclusions drawn, which are vaguely all-encompassing. About all that can be said with confidence is that changes in the erythron were consistent with an increase in oxygen-carrying capacity that is consistent with aerobic training, and that there were changes in cytokine profile that were reasonably consistent within the limits of the methodology employed and suggestive of a role for these markers of innate immunity in the training response. This is an observational study with a convenience sample and no controls. A designed experiment with dedicated experimental subjects and appropriate controls would be necessary to draw conclusions as firm as those offered here.

There remain significant issues with the use of language - language editing is required for clarity.

Reviewer 3 Report

Revised version looks much better; just a few minor text edits/grammar changes need to be made (e.g. Line 253 'hypothesises' change to 'hypothesized').

Author Response

The text was checked.